# Real-Time Fluorescence Visualization and Quantitation of Cell Growth and Death in Response to Treatment in 3D Collagen-Based Tumor Model

**DOI:** 10.3390/ijms23168837

**Published:** 2022-08-09

**Authors:** Ludmila M. Sencha, Olga E. Dobrynina, Anton D. Pospelov, Evgenii L. Guryev, Nina N. Peskova, Anna A. Brilkina, Elena I. Cherkasova, Irina V. Balalaeva

**Affiliations:** Institute of Biology and Biomedicine, Lobachevsky State University of Nizhny Novgorod, 23 Gagarin Ave., 603950 Nizhny Novgorod, Russia

**Keywords:** 3D in vitro tumor model, collagen hydrogel, fluorescence imaging, fluorescent proteins, cell growth curves, cisplatin, recombinant targeted toxin, genetically encoded sensor, apoptosis

## Abstract

The use of 3D in vitro tumor models has become a common trend in cancer biology studies as well as drug screening and preclinical testing of drug candidates. The transition from 2D to 3D matrix-based cell cultures requires modification of methods for assessing tumor growth. We propose the method for assessing the growth of tumor cells in a collagen hydrogel using macro-scale registration and quantification of the gel epi-fluorescence. The technique does not require gel destruction, can be used for real-time observation of fast (in seconds) cellular responses and demonstrates high agreement with cell counting approaches or measuring total DNA content. The potency of the method was proven in experiments aimed at testing cytotoxic activity of chemotherapeutic drug (cisplatin) and recombinant targeted toxin (DARPin-LoPE) against two different tumor cell lines genetically labelled with fluorescent proteins. Moreover, using fluorescent proteins with sensor properties allows registration of dynamic changes in cells’ metabolism, which was shown for the case of sensor of caspase 3 activity.

## 1. Introduction

The tumor microenvironment is currently considered as one of the most important factors in the formation of resistance of tumor cells to the effects of cytotoxic compounds. It is characterized by physicochemical properties different from normal tissues and includes stromal non-malignant cells and non-cellular components, such as extracellular matrix (ECM) [1]. The ECM serves not only as a scaffold for tissue organization, but also provides important biochemical and biomechanical signals that guide cell growth, survival, migration and differentiation; it also modulates vascular development and immune cells’ functioning [2]. Embedded into the ECM, cells interact with this macromolecular network through their surface receptors, such as integrins, discoidin domain receptors, cell surface proteoglycans and the hyaluronan receptor CD44 [3]. In this regard, cells integrate signals from the ECM that determine their functions and behavior. The processes of malignization and tumor progression are associated with significant changes in the chemical composition, spatial organization and functional characteristics of the matrix components. Tumor cells acquiring the ability to synthesize their own extracellular matrix proteins have a high metastatic potential [4]. At the same time, cell adhesion to the extracellular matrix can inactivate pro-apoptotic molecules, such as Bax, and induce the expression of several anti-apoptotic genes, including Bcl2, which contributes to cell survival [5]. Despite the general recognition of the essential role of the matrix in carcinogenesis, tumor development, metastasis and response to treatment, this area requires further scientific attention. It is necessary to take into account the chemical composition of the tumor tissue and peculiarities of the associated tumor microenvironment when developing treatment methods, since exposure within the tumor microenvironment can significantly increase the effectiveness of therapy [6,7]. In addition, it becomes obvious that the tumor progression and tumor cell phenotype could be affected by mechanical influences from the extracellular matrix [8,9]. It engages signaling pathways which are activated in response to mechanical stress, thereby rearranging the cell metabolism.

At the moment, the overwhelming number of antitumor drugs, as well as their dosages, are being tested in experiments using 2D monolayer cell cultures. However, it has been recognized that such models have a number of limitations, since they do not reflect the situation in vivo in terms of the characteristics of the native three-dimensional (3D) microenvironment of the tumor. Therefore, in recent years there has been a trend towards the use of more complex 3D in vitro models. Such models should bridge the gap between traditional monolayer cell cultures and animal models by simulating the key features of the native microenvironment. The microenvironment in solid tumors is fundamentally different from that in normal tissues. The peculiar features include physico-chemical conditions such as pH, oxygen tension and interstitial pressure, as well as gradients of nutrients, metabolites, and secreted signal molecules (growth factors, cytokines) [10]. Furthermore, the tumor microenvironment is characterized by the specific organization of the ECM, altered cell-cell interactions and intercellular signaling, and other factors derived from the particular 3D architectonics of the tumor [11]. It was demonstrated that cells in 3D culture morphologically and physiologically differ from cells in a two-dimensional model [12,13,14]. Cells cultured in 3D models also demonstrate the greatest resistance to antitumor drugs compared to 2D cultures [15,16,17,18]. Therefore, to reflect the interaction of cells with the matrix, as well as to take into account mechanical properties and spatial organization, matrices based on artificial or natural polymers are used. The benefit of using natural biomaterials for production of 3D tumor cultures is due to partial recapitulation of the chemical identity of the ECM contributing to cancer initiation and progression [19]. Biomaterials-based scaffolds have many active groups and spatial patterns recognized by cells; thus, they transduce signals, controlling cancer cell proliferation, motility and (de)differentiation, as well as formation and sprouting of capillaries [20].

As an extracellular scaffold, natural matrices based on alginate gel, hyaluronate, gelatin or collagen are particularly attractive. In addition, extracts of tumor cells with high biological activity, called Matrigel, are often used as a matrix source [21]. Each type of material has different characteristics, which makes it more suitable for the intended application. Thus, hydrogels of plant-derived alginate have demonstrated their applicability as scaffolding structures to preserve cell viability and function [22,23]. However, such hydrogels demonstrate low cell adhesion and weak mechanical characteristics of the scaffold, which leads to poor reproducibility of cell-matrix interactions of a solid tumor in vivo [24]. The other material of carbohydrate nature, hyaluronic acid, demonstrates several good properties, such as biodegradability and the ability to completely resorb. Reduced cell adhesion promotes the formation of cellular aggregates, which partially reproduces the three-dimensional architecture of the tumor and simulates cellular interactions [25,26]. Gelatin is also used as a popular protein-based scaffold for different properties such as water solubility, promoting cell adhesion and cheapness [27,28]. The low induction of inflammation and the ease of controlling the degree of crosslinking also contributes to the use of this material in biotechnology as a drug carrier [27]. At the same time, collagen as the most dominant natural component of the ECM in vivo has undoubted advantages [29,30]. From a biological point of view, collagen has low antigenicity, weak inflammatory response, biocompatibility and biodegradability, while the structure of the collagen matrix makes it suitable for simulating the microenvironment conditions of a solid tumor in in vitro culture [31]. At the cellular level, collagen participates in cell adhesion, intercellular and cell-matrix communication, which may provide a more relevant response of tumor cells to therapeutic agents [19]. To the present moment, collagen-based matrices have demonstrated promising possibilities for use as 3D cultural platforms for studying tumor progression and the response of tumor cells to various types of treatment [15,32,33,34].

The transition from 2D to 3D matrix-based cell cultures requires modification of methods for assessing tumor growth. The usual techniques of measuring cell viability and death are difficult to apply when it comes to three-dimensional models. Therefore, it is relevant to search for new informative methods, including those with high throughput (high-throughput screening methods), for example, to assess the effect of anti-tumor drugs. The technologies of genetic fluorescent labeling of tumor cells represent a very promising tool for the study of living systems and various cellular processes [35]. Such labeling combined with time-lapse observation allows obtaining a much larger amount of information compared to methods involving measurements at a single time point.

In this work, we propose the technique for quantitative assessment of the growth and death of tumor cells in a collagen hydrogel in real time using a fluorescence-based approach. The main benefit of the technique is an ability to perform long-term observations without destroying the hydrogel. The potency of the method was proven in experiments aimed at testing cytotoxic activity of chemotherapeutic and targeted drugs against two different tumor cell lines. Moreover, it has been demonstrated that using genetically encoded sensors allows registration of dynamic changes in cells’ metabolism, which was shown for the case of sensor of caspase 3 activity, the key apoptotic effector protease.

## 2. Results

### 2.1. Characterization of Tumor Cell Growth in 3D Collagen Hydrogels

To be able to detect cell growth in the 3D collagen-based culture, we chose two fluorescent cell lines cell of different origin. The first of them, SKOV3.ip-kat is a previously created human ovarian carcinoma with stable expression of far-red fluorescent protein TurboFP635, also known as Katushka [36]. The second line, A431-GFP is a human epidermoid carcinoma; the polyclonal cell line was obtained by stable transfection of parental A431 cells with GFP gene using plasmid vector and lipofection techniques.

The optimal number of the cells to be embedded in collagen hydrogels was established in a series of preliminary experiments, with the growth of cell culture without long lag-period or rapid degradation as the main criterion. Based on these experiments, collagen hydrogels with embedded SKOV3.ip-kat (6 × 10^5^ cells/gel) and A431-GFP (2 × 10^5^ cells/gel) cells were obtained. To measure the cell amount, we applied macro-scale epi-fluorescent imaging of collagen hydrogels with enclosed cells directly in culture plate wells (Figure 1) followed by quantitative analysis of the integral fluorescent signal. The autofluorescence of the gel itself obtained without cells’ encapsulation was measured as a control and subtracted from the measured values.

This approach made it possible to detect the minimal number of cells from the moment they were embedded in the gel. The fluorescent signal from hydrogels with SKOV3.ip-kat cells expressing the red fluorescent protein TurboFP635 visually differed from the control hydrogel without cells during the first day of the experiment, and then monotonically increased for 10 days (Figure 1A). The signal rise was attributed to the growth of cell numbers, which was supported by microscopic observation (Figure 1B). A uniform distribution of the fluorescent signal indicating a uniform distribution of cells over the area of the collagen gel was observed throughout the experiment.

Similarly, the registration of fluorescent signals from hydrogels with A431-GFP cells enabled detection of the relative amount of the cells from the first day of the experiment (Figure 1A). By the fifth day of the experiment, the phenomenon of hydrogel contraction becomes clearly noticeable (Appendix A). The phenomenon is well-known for cells of different origin cultured in collagen and results from the cells’ ability to destroy and/or remodel the matrix [37]. In our experiments, the hydrogel contraction by A431-GFP cells was accompanied by a redistribution of the level of the fluorescent signal over the gel area. On the 10th day of the experiment, the uniform distribution of the fluorescent signal changed to an uneven distribution, with the brightest areas on the periphery.

To prove whether the fluorescence analysis can be a reliable measure for cell culture growth, we compare it with two invasive well-recognized methods, namely cell counting after gel destruction and analysis of total DNA content. Growth curves were obtained for cell cultures by measuring all of these parameters (Appendix A); and the correlations between the noninvasive fluorescent method and invasive methods were calculated (Figure 2). The invasive techniques require gel destruction for every time point, so, the curves in these cases were obtained from many individual gels, while the fluorescence was measured in the same gels throughout all the experiment. Both fluorescent and non-fluorescent parental cell lines were used in these experiments, since no difference in growth parameters was detected by invasive methods.

In all the pairs of comparison, the agreement between the fluorescent quantitative imaging and the alternative recognized techniques was very high. The Pearson correlation coefficient for SKOV3.ip(kat) cells was 0.911 (*p* = 0.0007) when comparing fluorescence signal and cells number; and 0.908 (*p* = 0.0007) when comparing fluorescence signal and DNA content (Figure 2A,B). Similar results were obtained for the second A431(GFP) cells lines. In this case, the Pearson correlation coefficients were 0.956 (*p* < 0.0001) and 0.951 (*p* < 0.0001), respectively (Figure 2C,D).

From these results, the non-invasive fluorescence measurement is a reliable technique for analysis of cell culture growth in collagen hydrogel. We want to emphasize that the method significantly reduces the labor cost of the study and the amount of materials used.

### 2.2. Analysis of the Cytotoxicity of Cisplatin and DARPin-LoPE against Tumor Cells Cultivated in Monolayer Culture

We have analyzed the sensitivity of tumor cells of SKOV3.ip-kat and A431-GFP lines to cisplatin and the recombinant protein targeted toxin DARPin-LoPE. Cisplatin is a platinum-based drug widely used in antitumor therapy [38]. The mechanism of its action is based on a chemical bonding to DNA, which ultimately leads to a disturbance of DNA replication and transcription leading to cell cycle arrest and forcing cells to enter apoptosis. Both studied cell lines showed high sensitivity to this antitumor agent (Figure 3A,B). The viability of SKOV3.ip-kat and A431-GFP cells was less than 20–35% even at the minimum of studied concentration (2.1 μM). For comparison, the IC_50_ for the HeLa Kyoto cell line when treated with cisplatin is 8.3 μmol, which is several times higher than the results we obtained [39]. Such data indicate a high toxicity of cisplatin for the both studied cell lines.

The second of the agents tested is DARPin-LoPE, which is a recombinant antitumor toxin specific to the HER2 receptor [36,40]. The targeted properties of DARPin-LoPE are realized through the HER2-specific DARPin9.29 protein, and toxicity is realized due to the *Pseudomonas aeruginosa* exotoxin A fragment LoPE. This toxin specifically binds to the HER2 receptor on the tumor cell surface and after internalization and enzymatic processing causes the arrest of protein translation. The toxicity of DARPin-LoPE was analyzed against the SKOV3.ip-kat cell line which is characterized by overexpression of the HER2 receptor. The sensitivity of this cell line to targeted toxin was shown to be significantly higher compared to cisplatin—the viability of the cell culture decreased to 50% already at a concentration of 1 pM (Figure 3C). The extremely high sensitivity can be explained by the enzymatic nature of the toxin activity, namely ADP-ribosylation of elongation factor 2. We should note that almost identical viability values were obtained in a wide range of toxin concentrations, differing by more than 3 orders of magnitude. It can indicate the existence of a cell fraction resistant to this agent.

### 2.3. Analysis of the Cytotoxicity of Cisplatin and DARPin-LoPE against Tumor Cells Cultured in 3D Collagen Models

Using the proposed technique of fluorescence-based measuring of cell growth in collagen hydrogel, we analyzed the cell culture response to the treatment with the two studied cytotoxic agents. To this end, we obtained the cultures’ growth curves in the presence of different concentrations of toxic agents.

SKOV3.ip-kat line preserved its sensitivity to cisplatin. The initial growth of the fluorescence signal was replaced by inhibition after addition to the medium of cisplatin in all the studied concentrations, starting from 2.1 μM (Figure 4A). The degradation of the culture was dose-dependent (Figure 4B). The exposure to more than 83 μM of cisplatin leads to a decrease in the fluorescence to a “zero” level indicating total or nearly total death of cells in the hydrogel.

From the growth curves obtained by measuring the fluorescence of the gels without destroying them, we estimated the IC_50_ of cisplatin against SKOV3.ip-kat collagen-based culture. Because of variability of the initial signal level resulting from the dispersion of parameters in gel preparation, we used the ΔI_FL_, the difference of the fluorescence signal the analyzed time point (72 h in this particular case) from the signal of the same gel on the treatment start day (Figure 4B). For SKOV3.ip-kat cells in collagen hydrogel, the IC_50_ was about 8 μM. This value significantly exceeds IC_50_ for a monolayer culture (less than 2.1 μM) and indicates an increased resistance of SKOV3.ip-kat cells cultured in hydrogels (Figure 4B).

The same approach was applied for analysis of SKOV3.ip-kat cells response to treatment with the targeted toxin DARPin-LoPE (Figure 4C,D). Again, the dose dependence was observed, but, similar to the monolayer culture, the presence of a resistant cell population was observed. In the range of DARPin-LoPE concentrations from 1 pM to 1 nM, about 50% of cells retained their viability, and this proportion decreased only at significantly higher concentrations (Figure 4C,D).

The behavior of A431-GFP cells in collagen hydrogel differed greatly from that of SKOV3.ip-kat (Figure 5). When treated with cisplatin, these cells continue active growing even in the presence of high drug concentrations which were almost completely lethal in the monolayer culture. The blocking of culture growth was reached only at cisplatin concentrations of more than 80 μM, indicating a manifold increase in resistance in collagen microenvironment.

### 2.4. Study of Cell Death Mechanisms in 3D Models of Tumor Growth Using an Apoptosis Biosensor Casper3-GR

In addition to registering cell growth in a collagen hydrogel using fluorescence detection, we have tested an approach based on recording the fluorescence parameters of a genetically encoded sensor. The objective of this part of work was to get more information on A431-GFP response to cisplatin treatment. To do this, we used a cell line A431-Casper3GR expressing a genetically encoded apoptosis sensor of the Casper group, namely Casper3-GR [41] (Appendix A). This FRET-sensor contains two GFP-like proteins, green and red, linked by a DEVD sequence. Cleavage of the DEVD by caspase-3 results in loss of FRET and switching the sensor fluorescence from red to green. Thereby, in order to estimate FRET efficiency, we applied the ratiometric approach based on recording not the level of fluorescence, but the shape of the spectrum using spectrofluorimetric measurement by a plate reader.

Prior to the addition of cisplatin, the spectrum had two emission peaks that correspond to the peaks of the donor (green, at ~510 nm) and acceptor (red, at 580 nm) of the FRET pair and reproduced well the spectrum provided by the sensor manufacturer [41]. Addition of cisplatin to the cell medium to a final concentration of 100 μM (the concentration leading to culture growth arrest) caused the well-manifested change in the shape of spectrum of the fluorescent sensors (Figure 6A). The acceptor peak gradually disappeared and the donor peak rose. The dynamics of this process was estimated by the ratio of donor (500–530 nm) and acceptor (560–590 nm) signals I_green_/I_red_ (Figure 6B). A gradual increase in I_green_/I_red_ registered as early as 1 h after starting the cisplatin treatment, providing evidence of the activation of apoptosis and supporting the cytotoxic mechanism of cisplatin action on A431 derived cells.

## 3. Discussion

Currently, three-dimensional in vitro models are becoming increasingly popular for studying the molecular and cellular mechanisms of tumor growth [42,43,44]. The reason for this trend is the accumulation of knowledge about how much the phenotype of cancer cells depends on their microenvironment and specific conditions [45,46]. Such models are more relevant in relation to the complex structure of a constantly changing tumor in vivo by taking into account the interaction of cells with the matrix, mechanical properties, three-dimensional organization, etc. To date, a number of different materials (gelatin, alginate, etc.) are used as scaffolds for 3D models of tumor growth [23,28]. Imitation of the microenvironment of a native tumor is necessary to bridge the gap between in vitro and in vivo models, so the ideal scaffold should contain natural or biomimicking components. Due to the predominance of collagen in the extracellular matrix, it is often selected to produce scaffolds for 3D cultures. In this work, we have used collagen hydrogel to create a model of human ovarian carcinoma SKOV3.ip-kat and human epidermoid carcinoma A431-GFP. The cells were embedded and evenly distributed in a thickness of hydrogel which made it possible to imitate the three-dimensional organization of tumor tissue. This is especially important to take into account, since the cellular uptake and intracellular action of chemotherapeutic drugs are strongly influenced by the tumor microenvironment, including the mechanistic influence of the scaffold, intercellular interactions and cell-matrix interactions [11,19]. In addition, the proliferative activity of cells is important for the effect of most chemotherapeutic compounds, which can lead to increased resistance to antitumor agents [47]. The limitation of such 3D models is the need to use more sophisticated methodological approaches, in particular, when testing the responses of tumor cells to drugs. It should be noted that the vast majority of the used methods require the destruction of the gel structure, which makes the research costly and time-consuming and, ultimately, leads to an increase in the cost of both preclinical testing and the final medical product.

Fluorescence analysis methods are widely used when working with three-dimensional cultures. However, in almost all cases these are microscopy-based approaches with fixation and processing of samples, which excludes the possibility of observing the dynamic behavior of tumor cells. Such approaches allow visualizing intracellular structures [48], obtaining information about cell morphology [45], and also study of some physiological processes [46]. We have proposed a fluorescent method for assessing the growth of tumor cells in a collagen hydrogel, which is non-invasive and allows us to quantify the growth of tumor cells in a collagen hydrogel in real-time without destroying it. Importantly, the proposed approach based on macro-scale fluorescence detection has shown good consistency with other methods of analysis, which supports its high relevance.

The relationship between the three-dimensional structure of the tumor and its resistance to therapy is being actively studied [49,50,51]. We have tested the proposed fluorescence-based approach to analyze the peculiar features of the cellular response to the action of cytotoxic low- and high-molecular agents during cell growth in a collagen hydrogel. It has been shown that the cell lines SKOV3.ip-kat and A431-GFP cultured in a monolayer have high sensitivity to the low molecular weight agent cisplatin. It can be noted that the related line to SKOV3.ip-kat, namely SKOV3 in the ATCC international catalog, has the status of cisplatin resistance [52]. However, in our work, such resistance was not confirmed, which may be due to a more malignant phenotype of SKOV3.ip [53]. Cultivation in collagen hydrogel led to an increase in the resistance of both cell lines to cisplatin. The most interesting result from our point of view is the demonstrated multifold increase in the resistance of A431-GFP. Cisplatin concentrations leading to 50% inhibition of cultures’ growth in the collagen hydrogel differed by an order of magnitude (about 8 μM for SKOV3.ip-kat, and more than 80 μM for A431-GFP). Our data on the increased resistance of tumor cells cultured in 3D in vitro models compared to monolayer are in line with the results of other studies [54,55,56]. Such stability may be caused by interaction of extracellular matrix molecules with cell surface receptors. These receptors activate intracellular signaling pathways in cells, which lead to multiple physiological effects including enhancing the unlimited growth of tumor cells, resistance to deficit of growth factors, avoidance of apoptosis, enhanced angiogenesis and promoted invasion and metastasis [2,57].

A similar approach based on fluorescence registration was used to assess the toxicity of the recombinant toxin DARPin-LoPE against SKOV3.ip-kat cells characterized by overexpression of the HER2 receptor. Previously, the high activity of this toxin and the structurally similar DARPin-PE40 toxin against HER2-positive tumor cells of different lines was shown both in vitro and in vivo [58]. In our experiments, no significant difference was found in the response of SKOV3.ip-kat to the action of DARPin-LoPE in a monolayer and a three-dimensional culture. This effect requires further study, since for human ovarian adenocarcinoma cells, it was previously shown that cultivation in collagen hydrogel can increase resistance to the effects of antitumor agents [15].

The fluorescent approach also made it possible to register the fluorescence parameters of a genetically encoded sensor from cells embedded in a collagen hydrogel and proved itself to be a potent tool for registering the response of tumor cells to the action of a toxic agent in dynamics. In our work, this allowed us to reveal the apoptotic mechanism of cell death in response to cisplatin. We believe that the possibility of registering the fluorescence parameters of protein sensors provides huge opportunities for minimally invasive investigation of intra- and intercellular processes under conditions as close as possible to in vivo conditions. The wide range of currently available sensors enables registration of a wide variety of processes from the production of reactive oxygen species and Ca^2+^ signaling to the expression of genes of interest [59,60,61]. Despite the fact that in this study the proposed fluorescent approach was tested on a model of tumor cells embedded in a collagen hydrogel, there are no obstacles to the use of other materials as a scaffold. Change of the material can contribute to the ultimate goal of the researcher, while maintaining the ability to quickly and conveniently assess cell growth in a 3D model and obtain information about the peculiarities of cell metabolism in three-dimensional conditions.

As noted earlier, 3D models currently offer more and more opportunities for use as preclinical platforms for screening various candidate drugs and treatment methods and for obtaining a mechanistic understanding of the regulation of cancer cell death and viability in conditions that mimic the conditions of the tumor microenvironment. We believe that our proposed fluorescence-based approach can become a potent tool for research in this field, allowing monitoring the dynamics of the response of cells to various antitumor treatments and revealing the mechanisms underlying it.

## 4. Materials and Methods

### 4.1. Cell Culture

The work was carried out using the following cell lines: human ovarian carcinoma SKOV3.ip (provided from cell collection of the Institute of Bioorganic Chemistry of the Russian Academy of Sciences, Moscow, Russia); its derivative line SKOV3.ip-kat [36] stably expressing far-red fluorescent protein TurboFP635 [62]; human epidermoid carcinoma A431 (Russian collection of cell cultures, Moscow, Russia); and its derivative lines A431-GFP and A431-Casper3GR created as described in Section 4.2.

Cells were cultivated on Dulbecco’s Modified Eagle’s medium (DMEM, PanEco, Russia) containing 2 mM glutamine (PanEco, Moscow, Russia), 10% (*v*/*v*) fetal bovine serum (HyClone, Logan, UT, USA), 50 μg/mL penicillin and 50 μg/mL streptomycin (PanEco, Moscow, Russia) at 37 ℃ in 5% CO_2_ atmosphere. For passage, the cells were carefully detached with a trypsin-EDTA solution for cell lines A431, A431-GFP and A431-Casper3GR or Versen’s solution for cell lines SKOV3.ip and SKOV3.ip-kat. The change for Versen’s solution was due to the need to prevent proteolysis of membrane receptors including HER2 (PanEco, Moscow, Russia).

### 4.2. Stable Transfection of A431 Cells

The cells of A431 line were transfected with plasmid vectors encoding GFP (provided by Prof. B.S. Melnik, Institute of Protein Research of the Russian Academy of Sciences, Pushchino, Russia) or caspase3 sensitive sensor Casper3-GR [41]. Lipofection with Lipofectamin 3000 reagent (Invitrogen, Carlsbad, CA, USA) was performed according to the manufacturer’s instructions. The transfectants were preliminary selected on G418-containing medium followed by three repeated cycles of optical sort (FACS Aria III, BD Biosciences, San Jose, CA, USA) and expansions in cell culture to obtain the stably transfected cell population with bright fluorescence. For GFP fluorescence measurement, excitation at 488 nm and signal collection at 515–545 nm were used. In case of FRET-sensor Casper3-GR, fluorescence of both TagGFP donor (514–545 nm) and TagRFP acceptor (600–630) was measured with excitation at 488 nm.

### 4.3. Production of Collagen Hydrogel-Based 3D Tumor Model

A solution of collagen type I was first prepared according to [15] from rat tails and stored in sterile 0.1% acetic acid at a final collagen concentration of about 1.2 mg/mL at 4 °C. Type I collagen extracted from rat tails contains two different types of α chain and β-chains, with a total molecular weight of about 400 kDa [63] and isoelectric point in the range of pH 5.5–6.0 [64].

Sterile stock solutions were precooled to 4 °C in order to prevent fast gelation and used to prepare a mixture (Mix1) containing 10× DMEM (Biowest, Nuaillé, France), 25 mM glutamine (PanEco, Moscow, Russia), 1M HEPES (PanEco, Moscow, Russia), and 50% fetal bovine serum (HyClone, Logan, UT, USA).

Collagen hydrogels were obtained in separate wells of 6- or 12-well plates for tissue cultures (Corning, New York, NY, USA):To obtain hydrogels in a 6-well plate, 1600 μL of cooled collagen solution, 450 μL of Mix1, 2 × 10^5^ cells A431-GFP, or 6 × 10^5^ SKOV3.ip-kat, or 1.2 × 10^6^ A431-Casper3GR in 200 μL of DMEM, and 134 μL of 0.34 M NaOH were thoroughly mixed. The gels were incubated at 37 °C in 5% CO_2_ for 15–20 min until complete gelation. After solidification of the hydrogel, 2 mL of full serum-supplied DMEM without phenolic red (Gibco, New York, NY, USA) was added to the wells and the hydrogels were carefully separated from the walls of the wells with a pipette tip so that they were evenly surrounded by a medium.To obtain hydrogels in a 12-well plate, 800 μL of cooled collagen solution, 225 μL of Mix1, 2 × 10^5^ cells A431 or 2 × 10^5^ SKOV3.ip in 100 μL of DMEM, and 67 μL of 0.34 M NaOH were thoroughly mixed. The gels were incubated at 37 ℃ in 5% CO_2_ for 15–20 min until complete gelation. After solidification of the hydrogel, 1 mL of full serum-supplied DMEM was added to the wells and the hydrogels were separated from the walls of the plate wells.

The resulting hydrogels with embedded cells were incubated at 37 °C in 5% CO_2_; the growth medium was changed daily to a fresh medium.

### 4.4. Analysis of Tumor Cell Growth in Collagen Hydrogels by Cell Counting

To assess the rate of cell growth in a 3D model by cell counting, collagen hydrogels were subjected to enzymatic destruction. The hydrogel or its section was placed in a serum-free DMEM medium containing 0.08% collagenase type I (Gibco, New York, NY, USA) and 0.02% trypsin (PanEco, Moscow, Russia) for 1 h at 37 °C. The ratio of gel-to-media volumes was 5:8. After enzymatic destruction of hydrogels, 0.1 mL of 2% trypan blue solution (Sigma-Aldrich, Prague, Czech Republic) was added to the suspension of cells obtained, vortexed and after 5 min the number of live and dead cells was counted using a hemocytometer. To plot the growth curves of cell cultures in collagen hydrogels, the number of cells was counted in three separate gels every day for 10 days.

### 4.5. Evaluation of Tumor Cell Growth in Collagen Hydrogels Using Total DNA Measurement

The hydrogels were subjected to enzymatic destruction as described in Section 4.4. After obtaining the cell suspension, the cellular pellet was separated from hydrolyzed collagen by centrifugation at 400× *g* for 10 min. The total DNA was isolated using the ExtractDNA Blood DNA purification kit (VM011, Eurogen, Moscow, Russia) according to the manufacturer’s instructions. The concentration of dissolved DNA was measured on a NanoVue droplet spectrophotometer (GE Healthcare, Fairfield, CT, USA). To plot the growth curves of cell cultures in collagen hydrogels, the amount of total DNA was measured in three separate gels every day for 10 days.

### 4.6. Fluorescent-Based Measurement of Tumor Cells Growth in Collagen Hydrogels

Collagen hydrogels with embedded fluorescent cells were visualized daily on the DVS-03 fluorescent whole-body imager (Institute of Photonic Technologies of the Russian Academy of Sciences, Moscow, Russia). The images of the whole culture plate were obtained followed by analysis of the fluorescent signal in regions of interest (Appendix A). In the case of SKOV3.ip-kat cells, a LED source with a wavelength of 590 nm was used, and the signal was recorded in the range of 625–675 nm. In the case of A431-GFP cells, a LED of 490 nm and the signal recording within 513–556 nm were applied. Image capture time (exposure) in all cases did not exceed 10 s. Image processing was carried out using the ImageJ program (version 1.50i, National Institute of Health, Bethesda, MD, USA).

### 4.7. Production of Targeted Toxin DARPin-LoPE

DARPin-LoPE was produced in Escherichia coli BL21 (DE3) cells transformed with DARPin-LoPE-encoding plasmid and purified by metal-affinity chromatography, as described previously [32]. The DARPin-LoPE concentration was measured using BCA Assay Kit (Thermo Fischer Scientific, Waltham, MA, USA) according to the manufacturer’s instructions.

### 4.8. Cytotoxicity Assay of Cisplatin and DARPin-LoPE against Monolayer Culture and 3D Collagen-Based Model

The effects of cytotoxic drug cisplatin and targeted antitumor toxin DARPin-LoPE against monolayer culture were estimated using the microculture tetrazoline test (MTT). The cells were seeded in 96-well plates at the density of 2 × 10^3^ cells per well and allowed to attach overnight. The medium was then exchanged with fresh growth medium containing different concentrations of cisplatin (2.1–100 μM) or DARPin-LoPE (10^−3^–10^3^ nM) and the cells were incubated for 72 h. The medium was then changed to the fresh serum-free medium with 0.5 mg/mL MTT (3-(4,5-dimethylthiazol-2-yl)-2,5-diphenyltetrazolium bromide) (Alfa Aesar, Haverhill, MA, USA), and the cells were incubated for 4 h. Formazan formed from the reduction of MTT by mitochondrial dehydrogenases was dissolved in dimethyl sulfoxide (PanEco, Moscow, Russia), and the absorbance was measured at 570 nm with Synergy Mx plate reader (BioTeck, Vermont, VT, USA). Cell viability was calculated as a ratio (in percentage) of the optical density of treated to untreated cells. All results are expressed as mean ± standard deviation (SD). Data analysis and calculation of IC_50_ were performed using the GraphPad Prism software (GraphPad Software, version 6.0 for Windows, San Diego, CA, USA, 2012).

To measure the cytotoxic effect of the drugs against a 3D collagen-based model of tumor growth, the SKOV3.ip-kat and A431-GFP cells were embedded into hydrogels and grown for 72 h. Then, the medium was changed to the medium with 2.1–100 μM cisplatin or 10^−3^–10^3^ nM DARPin-LoPE and the hydrogels were further incubated for 72 h. The images of the hydrogels were obtained visualized daily and analyzed as described above in Section 4.6.

### 4.9. Registration of the Cell Apoptosis in Collagen Hydrogels

Development of apoptosis in response to cisplatin treatment was registered using A431-Casper3GR cell line. The prepared collagen hydrogels were left to grow overnight. The fluorescence spectra from hydrogels with embedded cells were recorded using plate reader Synergy Mx (BioTek, Vermont, VT, USA) using bottom-registration mode. The emission of Casper3-GR FRET-sensor was excited at 482 nm and recorded in the range of 500–700 nm (Gen5 software, BioTek, Winooski, VT, USA). The spectrum registered for ‘blank’ hydrogel without cells was subtracted for further processing. Then, the ratio I_green_/I_red_ was calculated by dividing the integral signals in the green region (500–530 nm, corresponds to donor fluorescence) and red region (560–590 nm, acceptor fluorescence).

## 5. Conclusions

We have proposed a method for assessing the growth of tumor cells in a collagen hydrogel using macro-scale registration and quantification of the gel epi-fluorescence. The technique does not require gel destruction, can be used for real-time observation of fast (in seconds) cellular responses and proved its reliability. Moreover, it allows significant cost reduction of cellular toxicity studies due to an ability to get the maximum information from the minimal quantity of individual hydrogels. We assume that the approach can be combined with the systems for the automated production and cultivation of 3D cell cultures, thus being applied for high throughput screening. An additional advantage can be obtained by combination of the method with genetically encoded sensors. In this case not only cell quantification but detailed analysis of their metabolic peculiarities can be performed.

## Figures and Tables

**Figure 1 ijms-23-08837-f001:**
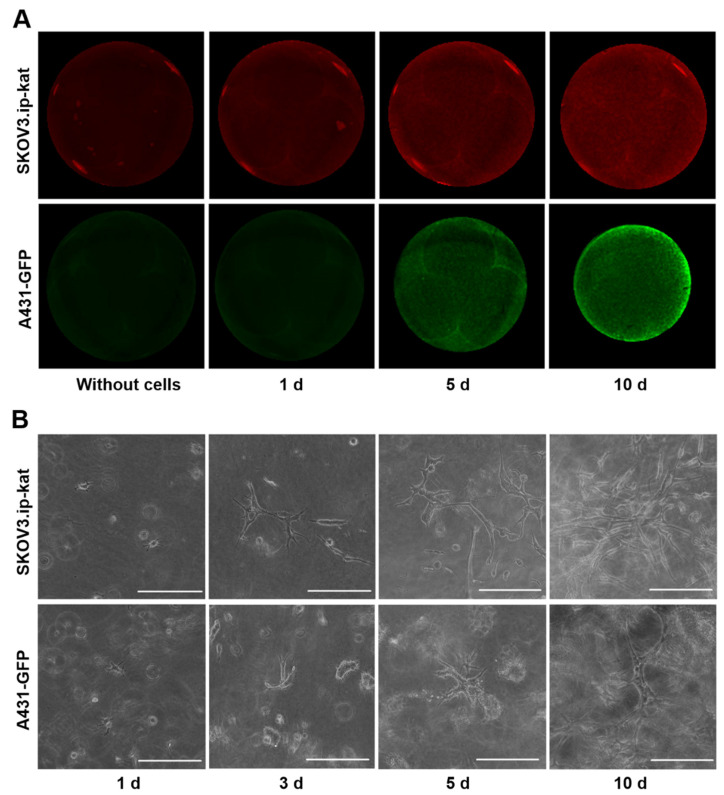
Growth of fluorescent cell lines SKOV3.ip-kat and A431-GFP in collagen hydrogels. (**A**) Epi-fluorescent images of control collagen hydrogel without cells and hydrogel with embedded tumor cells on days 1, 5, and 10 of the experiment. (**B**) Bright-field microscopic images of cell cultures in collagen hydrogel obtained on different days after the cells’ were embedded. Scale bar, 200 μm.

**Figure 2 ijms-23-08837-f002:**
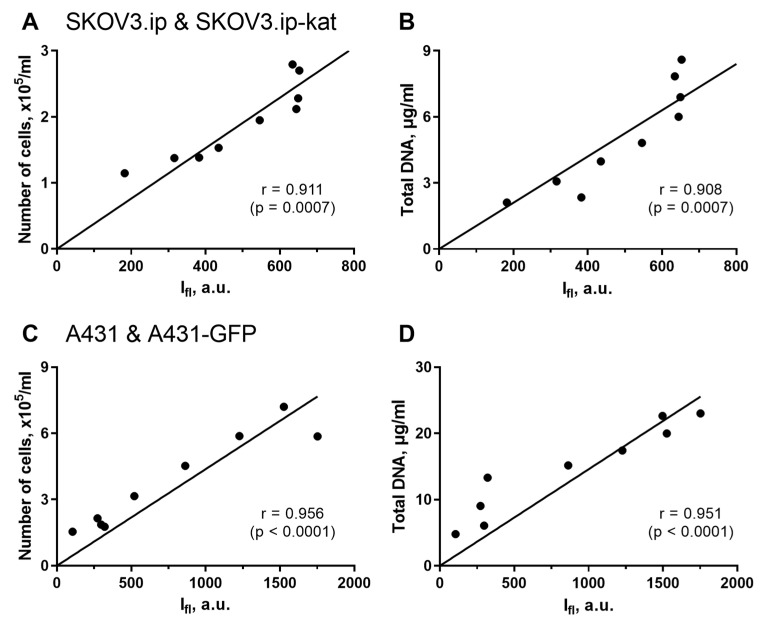
Analysis of correlation between fluorescence-based and invasive methods for assessing tumor cell growth in 3D collagen models. Regression analysis was performed for integral fluorescence intensity vs. number of cells (**A**,**C**); and for integral fluorescence intensity vs. total DNA content (**B**,**D**). The data for SKOV3.ip(kat) (**A**,**B**) and A431(GFP) (**C**,**D**) cells are presented. The Pearson correlation coefficients (r) with significance level (*p*) are indicated for all the comparisons.

**Figure 3 ijms-23-08837-f003:**
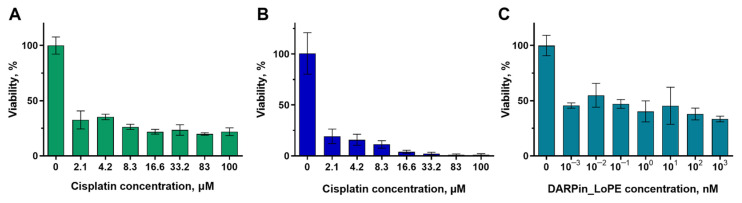
The dependence of the viability of monolayer cultures on the concentration of antitumor agent 72 h after the start of exposure. (**A**,**B**) Cisplatin toxicity against the SKOV3.ip-kat (**A**) and A431-GFP (**B**) cell lines. (**C**) Activity of recombinant targeted toxin DARPin-LoPE against the SKOV3.ip-kat cell line. The data are presented as mean ± standard deviation.

**Figure 4 ijms-23-08837-f004:**
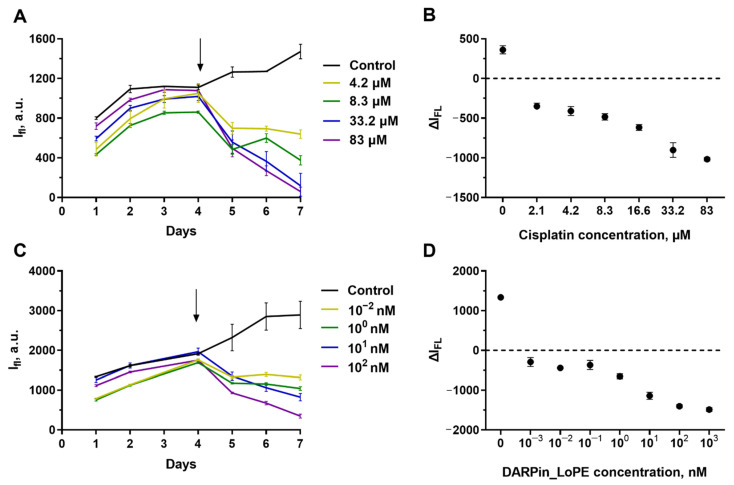
Cytotoxicity of cisplatin (**A**,**B**) and targeted toxin DARPin-LoPE (**C**,**D**) against SKOV3.ip-kat cells embedded in collagen hydrogel. (**A**,**C**) Dynamics of changes in the fluorescence signal from hydrogels treated with different concentrations of antitumor agents. On day ‘zero’, 6 × 10^5^ cells were embedded in the gel. The day cisplatin or DARPin-LoPE was added to the medium is indicated by an arrow. (**B**,**D**) ΔI_fl_ values 72 h after the start of hydrogels exposure to cisplatin (**B**) and DARPin-LoPE (**D**). The data are presented as mean ± standard deviation.

**Figure 5 ijms-23-08837-f005:**
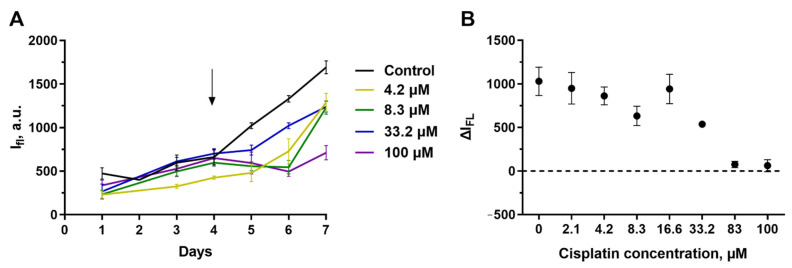
Cytotoxicity of cisplatin against A431-GFP cells embedded in collagen hydrogel. (**A**) Dynamics of changes in the fluorescence signal from hydrogels treated with different concentrations of antitumor agents. On day ‘zero’, 2 × 10^5^ cells were embedded in the gel. The day cisplatin was added to the medium is indicated by an arrow. (**B**) ΔI_FL_ values 72 h after the start of hydrogel’s exposure to cisplatin. The data are presented as mean ± standard deviation.

**Figure 6 ijms-23-08837-f006:**
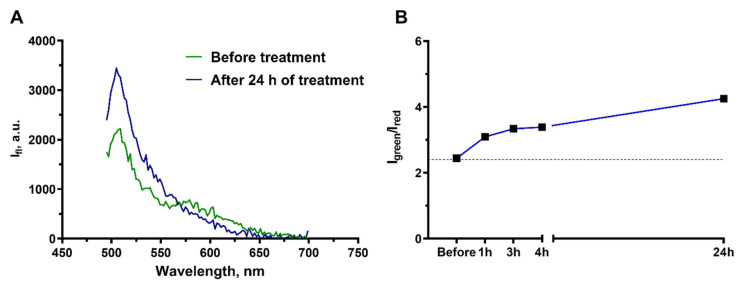
Monitoring of the apoptosis development in the culture of A431-Casper3GR cells treated with 100 μM cisplatin. (**A**) Spectra of fluorescence emission of collagen hydrogel with embedded A431-Casper3GR cells before and 24 h after cisplatin addition to the medium. (**B**) Change in the I_green_/I_red_ ratio of the Casper3-GR fluorescence at different time intervals after the addition of cisplatin.

## Data Availability

The data presented in this study are available on request from the corresponding authors.

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
