# Peer review of "Real-Time Fluorescence Visualization and Quantitation of Cell Growth and Death in Response to Treatment in 3D Collagen-Based Tumor Model"

_ijms, 2022, doi:10.3390/ijms23168837_

Round 1
Reviewer 1 Report
This research is interesting, but some descriptions or sentences are lacking, so it is difficult to understand the novelty. The authors should revise the manuscript based on the reviewer’s comments. At the current version, the reviewer cannot recommend the publication. Taken together, major revisions should be made before re-submission. The paper would be re-considered only when all comments were correctly responded.
1. Introduction
The relationship between 3D tumor models and natural biomaterial is not clear. The use of collagen gel is one of the most important points in this study. To understand the study, the authors should introduce why the biomaterial can assist the 3D cancer model. In lines 64-65, one sentence is but the description is poor. I recommend some recent review papers to be quoted.
Cancers 12 (10), 2754
http://doi.org/10.1089/ten.teb.2009.0676
Next, the authors should introduce the properties or merits of representative material for the 3D cancer model by quoting not only the research paper (the reference cited already is OK.) but also the review. Especially, the characteristics of materials are essential. Without the sentences, it is impossible to understand why the collagen was used. The short introduction is in lines 66-67, but the reviewer thinks it is not very understandable. I recommend some recent review papers to be quoted.
Gelatin
Molecules 2021, 26(22), 6795
Alginate
Microarrays 2015, 4(2), 133-161
HA
doi.org/10.1016/j.carbpol.2016.05.005
2.
4.3. Production of collagen hydrogel-based 3D tumor model
The detail of collagen should be mentioned, such as molecular weight or isoelectric point.
3. Figure 6
Some images using a microscope or timelapse are needed.
4. Figure 2
Coefficient determination should be added to claim the sensitivity.
5. Figure 3
The reviewer thinks the viability curve is more common.
6. Discussion
This part can not be understood. The authors should discuss the strength by quoting related papers above and comparing these researches.
Author Response
Dear Reviewer,
We would like to express our sincere appreciation for your careful attention to our manuscript and for the suggested improvements and valuable comments. We have revised the manuscript according to your remarks. Please see the attachment for a detailed description of the revision with our point-by-point answers.

Reviewer 2 Report
This manuscript describes a novel real‐time fluorescence method for measuring tumor cell growth in 3D and its inhibition by compounds used in chemotherapy. This technique will be of considerable interest to a large number of workers in the cancer field. It is well-written, aside from a few typos and misspellings that can easily be corrected.
Author Response
Dear Reviewer,
We would like to express our sincere gratitude for the interest to our work and the positive comments on the manuscript. We have checked the text for typos and misspellings.
Round 2
Reviewer 1 Report
I recommend the publication.